# Quantifying Firebrand and Radiative Heat Flux Risk on Structures in Mallee/Mulga-Dominated Wildland–Urban Interface: A Physics-Based Approach

Amila Wickramasinghe [1], Nazmul Khan [1], Alexander Filkov [2] and Khalid Moinuddin [1,*]

1 Institute for Sustainable Industries and Liveable Cities, Victoria University, Melbourne, VIC 3030, Australia; p.wickramasinghe@live.vu.edu.au (A.W.); nazmul.khan@vu.edu.au (N.K.)
2 School of Agriculture, Food and Ecosystem Sciences, Faculty of Science, University of Melbourne, Melbourne, VIC 3363, Australia; alexander.filkov@unimelb.edu.au
* Correspondence: khalid.moinuddin@vu.edu.au

**Abstract:** Fire spread in the Wildland–Urban Interface (WUI) can occur due to direct flame contact, convection, radiation, firebrand attack, or their combinations. Out of them, firebrand attack significantly contributes to damaging structures. To improve the resistance of buildings in wildfire-prone areas, the Australian Standards AS3959 provides construction requirements introducing Bushfire Attack Levels (BAL) based on quantified radiation heat flux. However, quantifying firebrand attack presents challenges, and the standard does not provide specific recommendations in this regard. This study aims to address this research gap by quantifying firebrand flux on houses according to the BALs in Mallee/Mulga-dominated vegetation using physics-based modelling. The study follows the AS3959 vegetation classifications and fire-weather conditions. The study considers Fire Danger Indices (FDI) of 100, 80, and 50 and identifies the housing components most susceptible to firebrand attack and radiant heat flux. The findings reveal an increasing firebrand flux with higher BAL values across all FDIs, with a greater percentage difference observed between FDIs 50 and 80 compared to FDIs 80 and 100. Furthermore, an exponential relationship is found between radiative heat flux and firebrand flux. This research contributes the development of effective strategies to mitigate the firebrand danger and enhance the resilience of structures to enhance AS3959.

**Keywords:** firebrands; physics-based modelling; AS3959; Wildland–Urban Interface; Malle/Mulga; Bushfire Attack Level; radiant heat flux





## 1. Introduction

Firebrands are identified as burning fragments of vegetation (twigs, leaves, bark, and cones) or construction materials [1,2] that can be lofted high into the atmosphere by fire-induced buoyancy and then transported by wind over distances ranging from a few metres to a few kilometres [3]. Once the firebrands land on a combustible fuel bed or a structure, they can start new fires (secondary ignitions) away from the primary fire [4]. Consequently, fire spread and structural ignition become harder to control, and firebrand attack contributes to the challenging and unpredictable nature of wildfire management [4,5].

Post-fire investigations conducted in 2003 of the Duffy community in Canberra, Australia, have provided compelling evidence that more than 50% of houses destroyed by wildfires were impacted solely by firebrands [6]. Moreover, investigations of the Witch–Guejito fire (USA) in 2007 revealed that two-thirds of houses ignited directly or indirectly from wind-dispersed firebrands [6]. Similar analyses conducted during several wildfires [7–10] in Australia have found that firebrands contributed to damage to over 90% of structures. The catastrophic Black Saturday bushfire of 2009 in Australia witnessed heavy firebrand attacks in the Kilmore East and Murrindindi regions, where a total of 1780 houses were lost due to fire events [11].

The dynamics of firebrands and landing locations exhibit substantial variability with fire-weather conditions, vegetation classification, firebrand morphologies, etc. The landing distance of firebrands that interact with the wind and the updraft flow depends on the magnitude of forces (gravity, weight, drag) exerted on them and the firebrand properties [12]. Structure ignitions can occur when these flaming or glowing particles land on housing components such as roofs, gutters, and decks [13], or penetrate houses through windows, gable vents, chimneys, and other vulnerable openings [13,14]. A comprehensive understanding of firebrand dynamics and quantitative assessment of firebrand attacks on structures under diverse fire and environmental conditions are important to formulate strategic plans to mitigate the vulnerabilities of houses in WUI.

Numerous standards for various jurisdictions propose guidelines to construct structures to mitigate the risk of firebrands. For instance, the FireSmart guidebook [15,16] in Canada, NFPA 1141, 1144 [15], the California Fire Code [17] in the US, the Fire Emergency New Zealand Act [18] and the New Zealand building code [19], the Forest Code of France [15], the fire code in Italy [20], AS3959 in Australia [21], and the International Wildland–Urban Interface Code [22] guide to mitigating the firebrand danger. These standards provide important information about buffer zones [15,16], defensible space between vegetation and houses, and construction requirements of vulnerable housing components [21].

Particularly in the AS3959 [21], the wildfire exposure to the houses is quantitatively explained in terms of Bushfire Attack Level (BAL)–radiative heat flux exposure thresholds. AS3959 considers seven vegetation classes, namely Forest, Mallee/Mulga, Scrub, Woodland, Shrubland, Rainforest, and Grassland. The severity of a fire is identified using the Fire Danger Index (FDI) (FDI 100, 80, 50, 40) [21] which varies according to the wind speed, ambient temperature, relative humidity, and the drought factor [23]. While AS3959 assesses the vulnerability of houses in relation to radiative heat flux, it does not offer a direct quantification of firebrand attack. The standard presents a qualitative observation that firebrand attacks may escalate with increasing Bushfire Attack Level (BAL). However, the inclusion of quantitative information about firebrand attacks in relation to the measurement of BALs is highly desirable to enhance the efficacy of this standard.

In our previous research [24], firebrand attacks on residential structures from Forest-classed vegetation were undertaken. The present investigation is centred on the application of the same method to the Mallee/Mulga fires. However, the methodology outlined in this investigation is applicable to other types of vegetation fires. It is important to underscore that this study introduces several novel dimensions, each significantly contributing distinctiveness from our prior work such as the calibration of firebrands tailoring them to the specific characteristics of Mallee/Mulga vegetation. Moreover, we quantify firebrand attacks from a considerably shorter vegetation compared to that in forests. This distinction is important as the fire's rate of spread is a function of the Malle/Mulga vegetation height and eventually links to the fire intensity that determines the firebrand-generation rate [25]. Additionally, we investigate the firebrand distribution probability in the downwind as a novelty of this work. Mallee/Mulga is a complex environment where wildfires are common [26].

Malle and Mulga are primarily found in arid (annual precipitation < 25 cm) and semi-arid regions in Australia. Mallee trees are usually short, growing from a single underground lignotuber (a woody structure at the tree base), and well adapted to low rainfall regions and able to survive in low-fertility soil [27,28]. Similarly, the Mulga vegetation type is also adapted to grow in harsh ecosystems such as in extreme temperatures and with limited water resources. The distribution of Mallee/Mulga vegetation in Australia and images of them are presented in Figure 1 [26]. This vegetation type predominantly extends across the southern and western parts of Australia, as well as in the southern part of the Northern Territory. Additionally, it can be found in certain areas of Victoria and New South Wales [26]. The expansion of human settlements to the outer skirts of the metropolitan areas has led to interactions within these Mallee/Mulga regions, potentially contributing

to Wildland–Urban Interface (WUI) fires. The Mallee/Mulga ecosystem, characterized by fire-prone evergreen sclerophyllous woodlands and shrublands, which exhibit higher flammability during dry summer months following the winter–spring growing season [29]. The wildfires in the Mallee ecosystem often reach high fire severity, expanding across approximately 1000 hectares due to the combustion of the upper parts of vegetation (top-kill) and the formation of large-scale, coarse-grained mosaics [30]. Unique burn-severity patterns distinguish Mallee wildfires from those in other ecosystems, such as temperate forests or tropical savannahs [30]. Critical fuel components in Mallee fires encompass leaf and bark litter [31], leading to fuel accumulation among trees and shrubs, [32] consequently forming distinct fuel patches. Observations indicate limited fire spread in Mallee vegetation on hot and dry days lacking strong winds [33]. However, once surpassing the fire-spread threshold, rapid propagation occurs, especially supported by strong winds prevalent in the Mallee/Mulga regions.

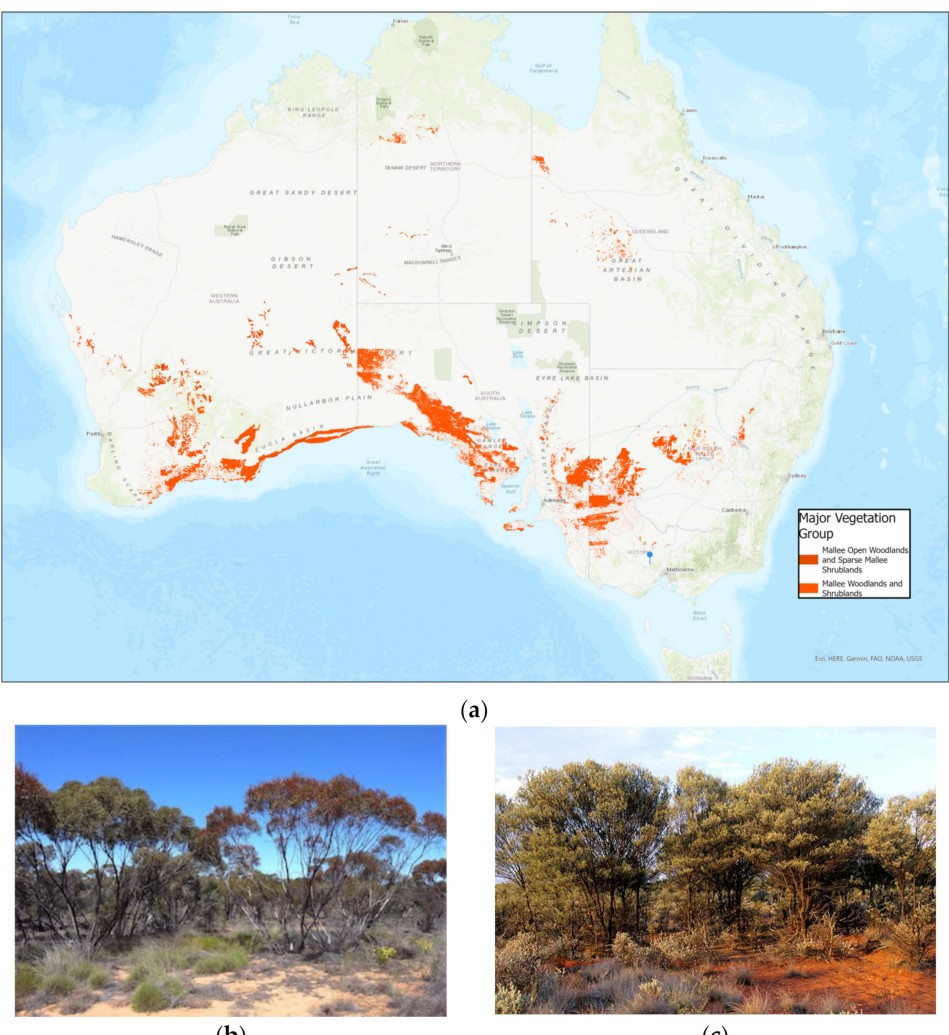

(**a**)

(**b**)                (**c**)

**Figure 1.** (**a**) Distribution of Mallee in Australia [26]; a picture of (**b**) Mallee vegetation [28] and (**c**) Mulga vegetation [34].

Therefore, it is important to quantify the firebrand attack within the WUI of the Mallee/Mulga regions while integrating the existing risk assessments outlined in AS3959 [21]. This paper focuses on fire events corresponding to Fire Danger Indices of 100, 80, and 50 that indicate the scale from severe to catastrophic fire conditions (as outlined in AS3959 according to Nobel et al. [23]). To achieve this, we employ a physics-based modelling approach. Physics-based modelling has emerged as a viable tool to simulate the key

processes of firebrand attack: generation, transport, landing, and secondary ignition, overcoming some of the limitations posed by safety, financial resources, and technology in field experiments [25,35–37]. In this study, we primarily focus on the following:

1. Creating atmospheric conditions for FDI 100, 80, and 50 for Mallee/Mulga regions.
2. Calibrating firebrand generation [25] in relation to the effects of wind speed, fuel moisture content (FMC), and vegetation species.
3. Simulating firebrand attack with radiative and convective heat fluxes on a modelled house including vulnerable housing elements.
4. Developing correlations between firebrand attack and radiative heat flux to propose AS3959 to enhance the safety of houses in the Mallee/Mulga WUI.

## 2. Methodology

A Fire Dynamics Simulator (FDS) [38] developed by the National Institute of Standards and Technology (NIST), USA, is used for the simulations. FDS is an open-source physics-based model that is widely used for fire research [25,35,36,39,40]. It solves the governing conservation equations of mass, momentum, and energy numerically in a three-dimensional space for low Mach number (Ma < 0.3) and thermally driven flow with an emphasis on smoke and heat transport from fires [38]. The FDS has the capability of simulating the dynamics of solid particles such as firebrands [25,35]. The fluid turbulence is accounted for by employing the Large Eddy Simulation (LES) methodology within the FDS-6.6.0 version using the Deardorff turbulence viscosity model [41]. Following Wadhwani et al. [36], we modified the FDS source code to incorporate the effect of sphericity of firebrand particles according to the Haider and Levenspiel drag model [42].

### 2.1. Simulation Domain

The simulation domain length was chosen as 336 m including the space needed for wind flow development as the open land (120 m), the length of the Mallee/Mulga vegetation region (130 m), the length of the house, and the gap to be maintained between the house and the vegetation proposed by AS3959 [21]. The width of the domain is 100 m, equivalent to the length of the fireline as per AS3959 [21], and the height is 60 m providing sufficient space for flame spread and firebrand movement. The domain is divided into 16 meshes parallel to the YZ plane as illustrated in Figure 2. Meshes 1 to 6 (green), on which wind development occurs, are discretised into 1.5 m cubic cells and meshes 7 to 16 (red), on which fireline and firebrand transport occur and are defined as 0.75 m cubic cells after a thorough mesh sensitivity analysis by Wickramasinghe et al. [25].

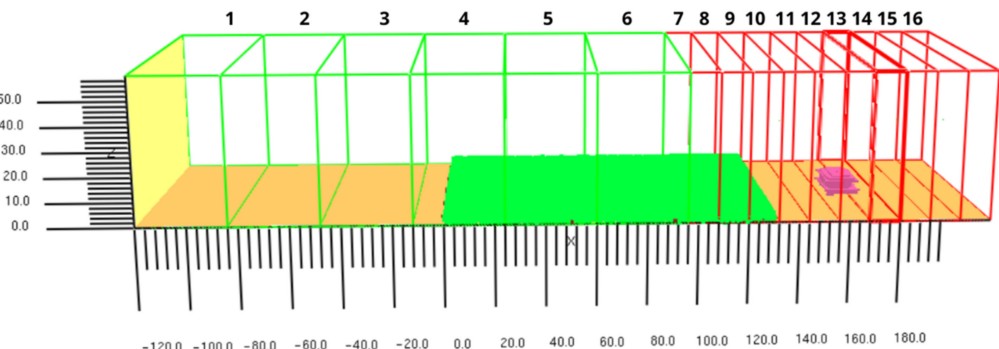

**Figure 2.** Schematic diagram of mesh arrangement and domain used in the Mallee/Mulga vegetation fire simulations. The green 1–6 meshes have a 1.5 m grid resolution, and the red 7–16 meshes have a 0.75 m grid resolution. The yellow wall at X = −120 m is the inlet of the domain where wind flow is applied. The green region at X = 0 to 130 m is the 3 m tall Mallee/Mulga vegetation, and the modelled house is shown in pink.

The inlet domain is at X = −120 m and the vegetation region (green) starts at X = 0 m providing sufficient open land space to develop the wind field. The end of the vegetation region is at X = 130 m. The left and right boundaries were set as 'MIRROR' boundary conditions (free slip) to avoid any edge effect while representing the extending nature of the fireline and vegetation. The 'OPEN' boundary condition is maintained at the top and the outlet boundaries of the domain. In our simulations, we focused on a flat landscape. Notably, AS3959 guidelines state that the distance (m) from the dominant vegetation class is consistent for both upward slopes and flat (0 degrees) lands [21]. However, the firebrand dynamics can be varied in slope conditions compared to the flat landscape. Due to the large number of simulations and limitations of the computational resources, we only conducted simulation for the flat land in this work.

### 2.2. Wind Flow Development Using the Atmospheric Profile

The methodology employed for developing the wind field is the wall of wind approach [43] by setting up the atmospheric profile. Following Jarrin et al. [44], the characteristic length scale and the count of turbulent eddies at the inlet were calculated. In this particular instance, the length scale of the eddies was determined as three times the cell length of the inlet domain (3 × 1.5 m), and the number of eddies (≈300) was calculated by dividing the area of the inlet domain by the square of the length scale ($(102\ m \times 60\ m)/(4.5\ m)^2$). The wind velocity $U_{10}$ is measured at X = −80 m at the open land to avoid the effect of turbulence forming at the front edge (X = 0 m) of the vegetation region. For each FDI, the inlet velocity is controlled to obtain the desired wind velocity $U_{10}$ at X = −80 m and 10 m elevation. We used three different steady-state precursor wind flows to feed in the actual simulations to obtain the developed wind fields in a shorter computational time. These precursor wind flows were created as a trial-and-error process without initiating fire or firebrands and running for 5 times (spin-up time) the domain travel time (5DTT) as per Moinuddin et al. [45]. The wind velocities at the domain inlet are maintained as 20.7, 18.2, and 14.3 m/s to obtain the 19.44, 16.67, and 11.11 m/s velocities at X = −80 m that also replicate the FDI 100, 80, and 50 fire-weather conditions as per Equation (1) [23]:

$$FDI = 2.0 \times e^{(-0.450 + 0.987 \times \ln(D) - 0.0345 \times Rh + 0.0338 \times T + 0.0234 \times U_{10})} \tag{1}$$

where $D$ is the drought factor, $Rh$ is the relative humidity (%), $T$ is the temperature (°C), and $U_{10}$ in km/h. Here, the wind speed is varied, while other parameters are kept as constants. The drought factor was taken as 10 which indicates the complete dryness of the fuel; the temperature was taken as 39 °C to match the annual maximum mean temperature in Australia [46]. The relative humidity was taken as 25% from the psychrometric chart ensuring the possibility of the existence of an extremely dry day according to the chosen ambient temperature [46]. The historical weather data provided by the Australian Bureau of Meteorology indicates the possibility of specific regions experiencing conditions where the relative humidity is 25% [46] while simultaneously having a maximum temperature of 39 °C [47]. Taking account of these conditions, the developed wind fields for FDI 100, 80, and 50 are presented in Figure 3. The wind velocity profiles over the open land with ≈10 m gap against the vertical (z) direction are plotted to observe the development of the flow before initiating the fire and firebrand generation at the fireline.

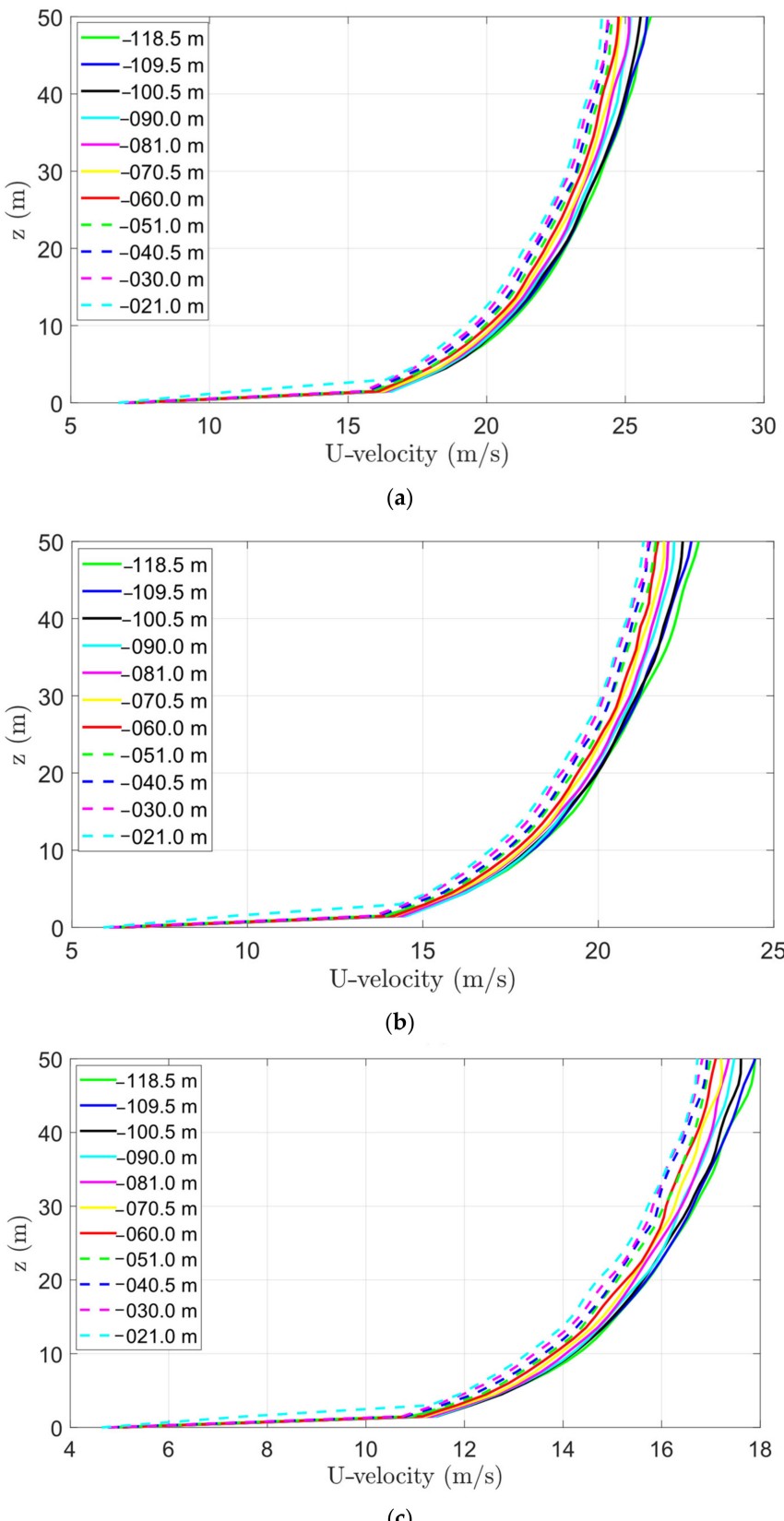

**Figure 3.** Wind flow development (*U*-velocity) of Malle/Mulga vegetation simulations for (**a**) FDI 100, (**b**) FDI 80, and (**c**) FDI 50. The *U*-velocity slice files are processed using MATLAB to create the profiles.

### 2.3. Vegetation

Mallee/Mulga is a multi-stemmed habitat dominated by *Acacias* shrubs with an average height greater than 2 m and foliage cover of <30%. It has an understorey consisting of dense low shrubs or sparse grasses [21]. As per AS3959, we cannot distinguish in our modelling the canopy and understorey in Mallee/Mulga vegetation. Therefore, we decided to spread fuel particles randomly throughout a layer of 3 m height from ground level with the respective total fuel loads of 8 t/h.

Depending on the shapes and sizes of tree leaves, the drag force exerted on wind flow by the vegetation varies. Therefore, representing reasonable shapes and sizes to match with actual tree leaves is important in these simulations. We selected the Acacia aneura species [48] which is widely distributed throughout all mainland estates of Australia [48] to determine the shapes and sizes of the tree leaves to represent Mallee/Mulga vegetation classification. The average length of tree leaves of the *Acacia aneura* is around 3–11 cm. [48]. The fuel density mass per unit volume $MPUV_{Mallee/Mulga}$ was calculated as 0.267 kg/m to use as a simulation input. These fuel particles were randomly spread over the volume of 130 m $\times$ 100 m $\times$ 3 m to represent the Mallee/Mulga vegetation layer as shown in Figure 2. The tree trunks are represented by non-burning solid obstacles with a height of 1.5 m and 0.75 m diameter (due to the 0.75 m cell size). While Mallee/Mulga naturally exhibits multi-stem growth, we simplified our modelling approach by representing them as single-stem vegetation. This simplification was required by the challenge associated with accurately modelling the multi-stem emanating from a common tree base as per the constraint imposed by the chosen grid resolution.

### 2.4. House Design

In AS3959, design requirements for various housing components are provided to minimize the effects of radiative heat flux (quantitatively) and firebrands (qualitatively). Therefore, having a compliant house design with relevant components such as a roof, gutters, sub-floor, decks, windows, doors, and steps is important in this work. We included a single-storey house in the FDS simulation with all these features following standard house designs proposed by the Australian government [49]. To measure the radiative heat flux, convective heat flux, and firebrand landing mass on the housing components, we set up devices that record these parameters as time progresses. The frontmost location of the house is positioned parallel to the front edge of the Mallee/Mulga vegetation region. The distance between the house and the vegetation is maintained according to the BALs given in Table 1. Notably, the proposed gaps for identical BALs remained unchanged for FDIs in Mallee/Mulga vegetation.

**Table 1.** Proposed AS3959 distance between the house and the Mallee/Mulga vegetation in [21] based on BALs.

| Case | BAL | | | | |
|---|---|---|---|---|---|
| | **BAL FZ** | **BAL 40** | **BAL 29** | **BAL 19** | **BAL 12.5** |
| | **Distance between the House and Vegetation (m)** | | | | |
| Mallee/Mulga FDI 100 | <6 | 6 – <8 | 8 – < 12 | 12 – < 17 | 17 – <100 |
| Mallee/Mulga FDI 80 | <6 | 6 – <8 | 8 – < 12 | 12 – < 17 | 17 – <100 |
| Mallee/Mulga FDI 50 | <6 | 6 – <8 | 8 – < 12 | 12 – < 17 | 17 – <100 |

According to Table 1, it is identified that a house is at the flame zone (FZ) when the distance is less than 6 m from the vegetation. Similarly, a house receives 40 kW/m$^2$ radiative heat flux (BAL 40) when it is at a 6 to 8 m distance from the vegetation. The house should be located between 17 and 100 m from the vegetation in order to receive 12.5 kW/m$^2$ radiative flux (BAL 12.5). We normally took the middle value of the given range of each BAL to position the frontmost location of the model house. However, for BAL 12.5, the

house was placed at 50 m which satisfies the condition outlined in Table 1 alongside aiding in decreasing the domain length, thereby reducing computational expenses.

### 2.5. Firebrand-Generation Rate

Due to the scarcity of data, the morphologies (size and shape) of firebrands were assumed to be similar to the firebrands collected from a management scale prescribed fire experiment conducted in New Jersey, USA, by Thomas et al. [50]. The images of collected firebrands were processed to identify the most suitable shape (cylinder, sphere, cube) and measured dimensions such as length, width, thickness, and diameter. These measurements were taken as inputs in the FDS simulation.

The amount of firebrand generation varies with fuel species [51–53], wind speed [54], and fuel moisture content, FMC [52,53,55]. Hence, accounting for these factors for estimating the firebrand generation is crucial. However, we could not identify any single experiment to quantify all effects of fuel species, FMC, and wind speed for the firebrand generation. However, some researchers have burnt different tree species under various FMC and wind speeds to find the firebrand generation. Therefore, we employed the experimental data of Hudson et al. [53] and Bahrani et al. [54] to develop correlations to quantify the effects of fuel species, wind speed, and fuel moisture content for firebrand generation. Owing to the scarcity of data, we assumed the tree species from the same family or with similar physical features produce a similar number of firebrands under the same environmental conditions. We compared the average tree height, the shape of the canopy, and the orientation of tree leaves of these species to identify the vegetation from the same family using photographs and databases [56,57] of numerous vegetation species. Out of the fuel species burnt by Hudson et al. [53], *Western Juniper* is the fuel species mostly matched with *Acacia*. Similarly, *Leyland Cypress* is the fuel species matching with *Acacia* among the trees burnt by Bahrani et al. [54] to find the effect of wind speed on firebrand generation. Using these experimental data, we derived mathematical relationships given by Equations (2) and (3) to find the firebrand generation of Mallee/Mulga (*Acacia*) in different wind speeds and FMC, respectively.

$$fb_{generation} = 62.05 \times U_{wind} + 475.5 \tag{2}$$

$$fb_{generation} = -28.63 \times FMC + 1688.8 \tag{3}$$

Here in (2) and (3), $fb_{generation}$ is the number of firebrand generation (pieces [pcs] from a single tree), and $U_{wind}$ is the wind speed (m/s). According to these correlations, the firebrand-generation number increases with the increase of wind speed and decreases with the increase of FMC.

It is noteworthy that with the entering of wind into the vegetation area, its speed reduces due to the drag force exerted. It is important to find the wind speed at the fireline (firebrand-generation region). For the open land wind speeds of 19.44 (FDI 100), 16.67 (FDI 80), and 11.11 m/s (FDI 50), we found from the test simulations that the average wind speeds at the fireline are 10.38, 8.88, and 5.48 m/s. These wind speeds are used with the developed correlation given in Equation (2) to estimate the firebrand generation for Mallee/Mulga.

To calibrate the firebrand-generation rate in terms of the heat-release rate, we used the firebrand-generation rate (4.18 pcs/MW/s) as the baseline estimated by Wickramasinghe et al. [25]. This estimation was attributed for the *Pitch Pine* vegetation fire in 2 m/s average wind speed and 31% FMC. From this base, we calculated the firebrand-generation numbers for the wind speeds (10.388, 8.88. and 5.48 m/s) at the fireline and 3.84% FMC, as shown in Table 2. Secondly, we computed the firebrand-generation ratios relative to the base conditions (2 m/s wind speed and 31% FMC) for the wind speeds and FMCs that replicate the FDIs of 100, 80, and 50. The use of the firebrand-generation ratio helps to quantify the increase or decrease of the amount of firebrand generation because of the changes in wind speed and FMC relative to the base conditions. The firebrand-generation rates related to

each FDI are calculated by multiplying the base firebrand-generation rate (4.18 pcs/MW/s) by the computed firebrand-generation ratios as shown in Table 2.

**Table 2.** Firebrand-generation rates at FDI 100, 80, and 50 for *Acacia* (Mallee/Mulga) vegetation.

| Similar Vegetations | Firebrand-Generation Rate (pcs/MW/s) | Wind Speed (m/s) | Number of Firebrands (pcs) | Generation Ratio | Generation Rate (pcs/MW/s) |
|---|---|---|---|---|---|
| *Leyland Cypress* and *Acacia* | 4.18 at 2 m/s wind speed (base information) | 2<br>5.48 (FDI50)<br>8.88 (FDI80)<br>10.38 (FDI100) | 599<br>815<br>1026<br>1119 | (599/599) = 1.00<br>(815/599) = 1.36<br>(1026/599) = 1.71<br>(1119/599) = 1.87 | 4.18 × 1.00 = 4.18<br>4.18 × 1.36 = 5.69<br>4.18 × 1.71 = 7.16<br>4.18 × 1.87 = 7.81 |
| **Similar Vegetations** | **Firebrand-Generation Rate (pcs/MW/s)** | **FMC (%)** | **Number of Firebrands (pcs)** | **Generation Ratio** | **Generation Rate (pcs/MW/s)** |
| *Western Juniper* and *Acacia* | 4.18 at 31% FMC (base information) | 31<br>3.84 | 801<br>1579 | (801/801) = 1.00<br>(1579/801) = 1.97 | 4.18 × 1.00 = 4.18<br>4.18 × 1.97 = 8.24 |

Hudson et al. [53] burned *Ponderosa Pine* and *Western Juniper* trees that have relatively similar mass and height under the experimental series conducted to find the firebrand generation according to fuel species. The firebrand-generation ratio between these two species is around 0.9. We assumed *Ponderosa Pine* and *Pitch Pine* produce a similar number of firebrands based on their physical appearance (tree height, canopy shape, tree needles), same family (Pinaceae), Genus (Pinus), subgenus (P. subg. Pinus), and same section (P. sect. Trifoliae). Among the tree species burnt [36], *Western Juniper* is the most similar species to *Acacia*. As we know the firebrand-generation rate of *Pitch pine* [25], we calculate the firebrand-generation rate for *Acacia* (Mallee/Mulga vegetation classification) as 4.18 pcs/MW/s × 0.9 = 3.76 pcs/MW/s. The total generation rates accounting for the wind speed, FMC species, and fuel species are found by multiplying the individual generation ratios and the firebrand-generation rate of *Pitch pine* for each severity of fire as shown in Table 3.

**Table 3.** Total-firebrand-generation rates for FDI 50, 80, 100.

| FDI | Base Firebrand-Generation Rate (pcs/MW/s) | Generation Ratio of Wind | Generation Ratio of FMC | Generation Ratio of Species | Total-Firebrand-Generation Rate (pcs/MW/s) |
|---|---|---|---|---|---|
| 50 | | 1.36 | | | 4.18 × (1.36 × 1.97 × 0.9) = 10.09 |
| 80 | 4.18 | 1.71 | 1.97 | 0.9 | 4.18 × (1.71 × 1.97 × 0.9) = 12.7 |
| 100 | | 1.87 | | | 4.18 × (1.87 × 1.97 × 0.9) = 13.85 |

These total-firebrand-generation rates are used to find the firebrand input rate (pcs/s) from the fire in the simulation. This was conducted by multiplying the total-firebrand-generation rates by the fireline's HRR calculated according to Equation (4) for each FDI:

$$HRR = I \times L \tag{4}$$

where *L* is the fireline length (100 m) and *I* is the fireline intensity given using

$$I = HoC \times W \times RoS \tag{5}$$

where *HoC* is the heat of combustion (18,600 J/kg) [21], *W* is the total fuel load (8 t/ha), and *RoS* is the rate of spread of fire (km/h) given using

$$RoS = 0.023 \times (U_{10})^{1.21} \times VH^{0.54} \tag{6}$$

where *VH* is the average height of the classified vegetation (3 m). The calculated firebrand input rates were used to initiate firebrands randomly over the spatial region (3 m × 100 m × fireline depth (m)) which is engulfed by the fire.

### 2.6. Determining Fireline Location

Setting the fire front at the edge of the Mallee/Mulga resulted in the flame leaning towards the house and engulfing it at some of the higher BALs such as BAL FZ. When this happens, the heat flux at the house is extremely high, and the nature of the flame is unrealistic. To avoid this, we brought the fire front 5 m back from the vegetation edge for Mallee/Mulga. Owing to this modification, the flame became more realistic, and the house did not catch fire. Moreover, we compared these radiative heat fluxes with the proposed values using the empirical model [21] given in AS3959. The comparison of the radiative heat fluxes of FDS simulations (with a 5 m gap between the vegetation edge and fire front) and the empirical model are presented in Figure 4.

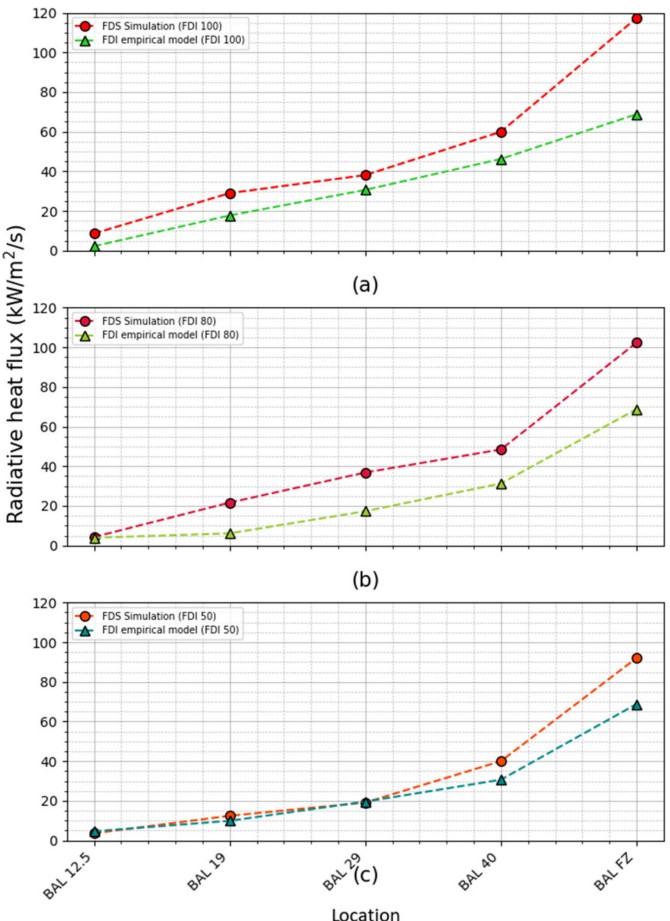

**Figure 4.** Heat flux from numerical simulations and the AS3959 empirical model for (**a**) FDI 100, (**b**) FDI 80, and (**c**) FDI 50 on the house from BAL 12.5 to BAL FZ. The distance between the vegetation edge and the fire front is 5 m.

Both the simulation results and the radiative heat fluxes calculated using the empirical model follow the same trend of increasing BALs for all the FDIs. As well as this, they are reasonably matched with the equivalent BALs especially for lower BALs such as BAL 12.5 to 29. The deviation is higher for BAL FZ. Overall, the location of the fireline satisfies the fact that the house receives the radiative heat flux given using the empirical model in AS3959 [21] which indicates a similar nature of modelled and given fire conditions.

### 2.7. Numerical Simulation

A total of 15 simulations were conducted for Malle/Mulga vegetation varying the five BALs for each of the three FDIs. The pre-computed wind fields (velocity CSV files) are loaded into the simulations depending on which FDI is considered. The simulations are run

for another 110 s to establish the imported wind profiles while observing their stability. The fire started at $t = 110$ s and 10 s post-ignition firebrand generation commenced to avoid the initial shock from starting the fire. Firebrand input into the domain is continued for 14 s to match the residence time of the fireline, which is calculated according to Morvan et al. [58].

The total simulation time is set to $t = 220$ s to provide sufficient time for the firebrands to land on the house and the ground. The first and the last arrival times of firebrands on the house and each strategic location (roof, gutter, deck, etc.) are recorded. The mass flux on the ground is also obtained in the downwind direction. The mass of firebrands moving through the vertical plane from $X = 132$ to 210 m is also recorded to understand the firebrand distribution in each case. Similarly, heat flux is also computed at the house.

## 3. Results and Discussion

### 3.1. Firebrand Landing Distribution

The firebrand distribution for each FDI is plotted against distance, as presented in Figure 5. The firebrand mass flux through the vertical planes (Figure 5a) shows an exponential relationship for all FDIs with $R^2 > 0.91$. The number of firebrands that pass through the vertical planes close to the vegetation edge is higher and decreases with distance. Additionally, we can observe that the particle flux decreases with FDI.

The firebrand landing mass flux downwind is presented in Figure 5b. Here, we calculated the total mass of firebrands' land per unit time on 6 m wide planes/portions on the ground in the positive $X$ direction starting from $X = 132$ m (edge of the vegetation). Similar to the flux through vertical planes, the firebrand landing mass flux is significantly higher at the horizontal planes near the vegetation. This rapidly decreases with distance, and after $X = 180$ m the particle landing is almost zero.

The firebrand landing distance in the X direction and the coloured contours are presented in Figure 6 along with the probability distribution. In this analysis, we considered only the particles that passed through the horizontal plane at z = 0.75 m from the ground level (1 cell thickness). It does not account for the particles that landed on the roof and the gutter area (upper region) of the house. Taking the BAL 29 (middle value of BAL scale) as an example, it shows that the accumulation of firebrands closer to the firefront is higher in FDI 50 than the FDI 80 and 100. The probability of firebrand landing also shows to be the same where the peak is higher in FDI 50 and becomes flatter with increasing FDI 80 and 100. This could be due to the lower wind speed not carrying the firebrands to distance in the downwind direction for lower FDIs.

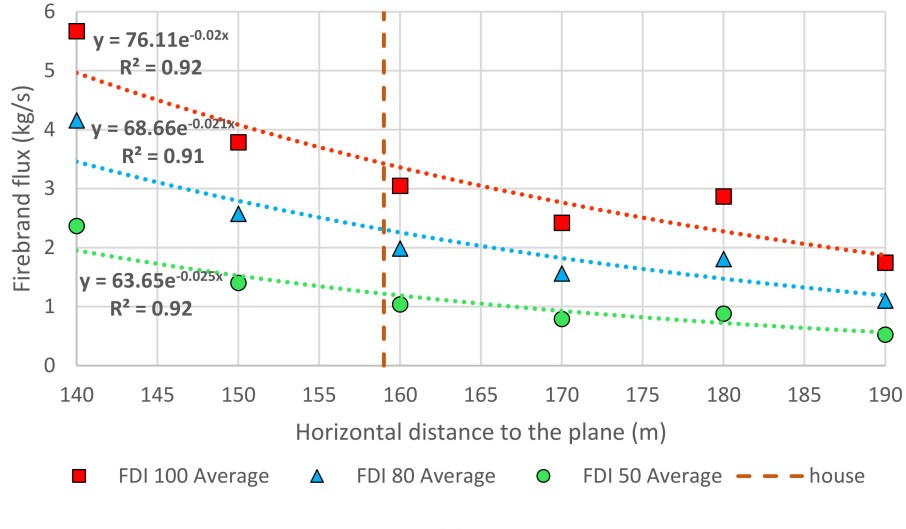

(a)

**Figure 5.** *Cont*.

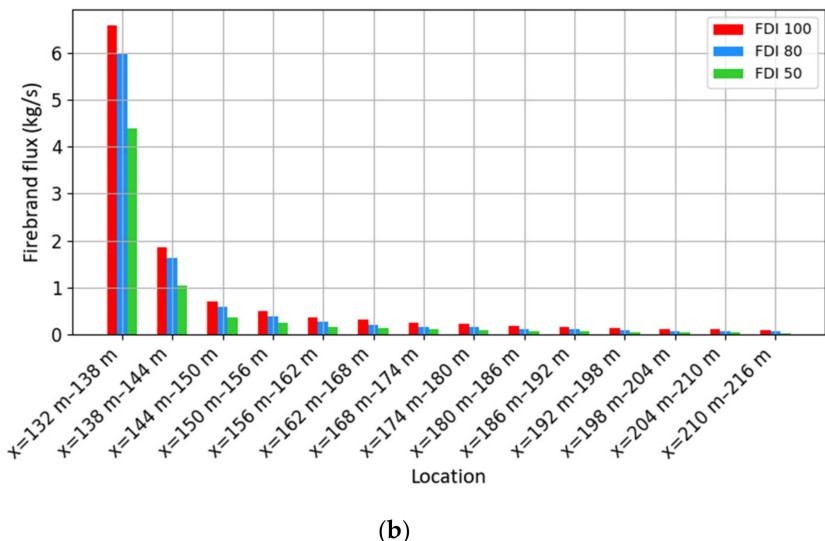

(**b**)

**Figure 5.** Average firebrand mass flux through (**a**) vertical planes, and (**b**) horizontal planes to examine firebrand transport towards the house at different FDIs in Mallee/Mulga vegetation fires.

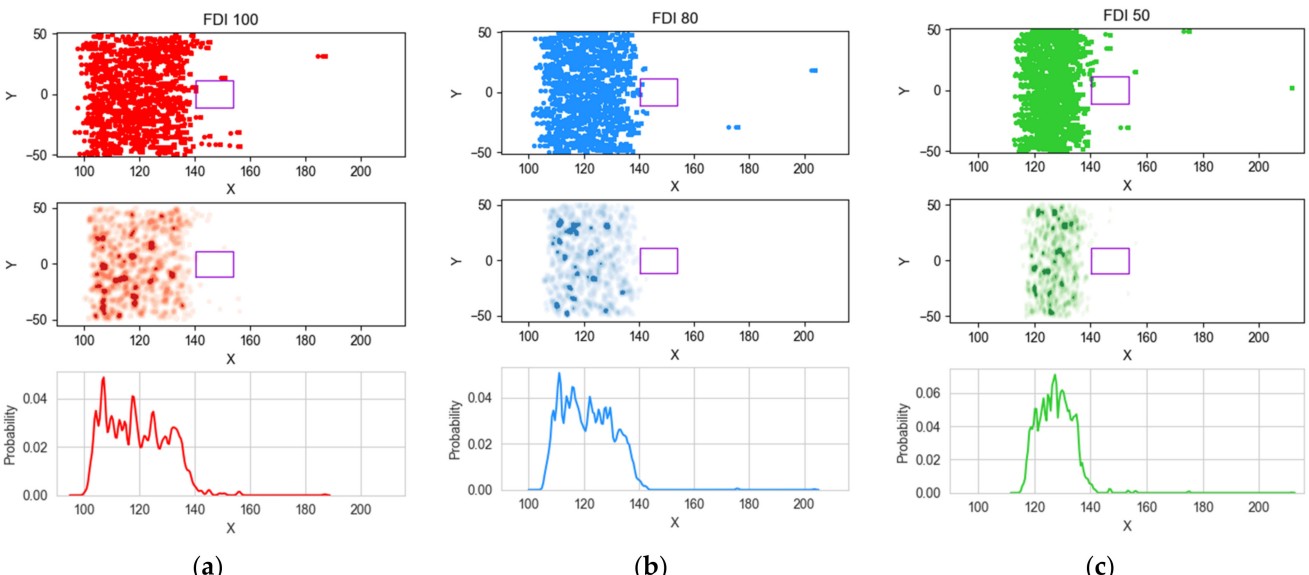

(**a**)                        (**b**)                        (**c**)

**Figure 6.** The plan view (XY plane) of firebrand landing locations (top), coloured contours of firebrand landing density (middle), and the probability distribution in XZ plane (bottom) for (**a**) FDI 100 (red), (**b**) FDI 80 (blue), and (**c**) FDI 50 (green) in BAL 29. The rectangle area (purple) is the location of the house.

### 3.2. Firebrand Flux on the House

In Figure 7, the effect of the firebrand flux on the house (first tick on *X*-axis) at FDI 100 exhibits a notable decline of 50%, 27%, 42%, and 56% when transitioning from BAL FZ to BAL 12.5. With the exception of the deck area in BAL FZ, FDS simulations reveal that none of the fire events results in a firebrand flux exceeding 0.5 pcs/m²/s. The deck and stairs, which are the frontmost locations of the house and are covered by the roof and eaves, experience comparatively higher firebrand flux. These locations are susceptible to firebrands with flat trajectories, which can penetrate into the internal part of the house. The lower average height of Mallee/Mulga vegetation (3 m) does not facilitate the release of firebrands from higher elevations. This phenomenon results in causing particles to predominantly follow flat trajectories, as is evident in the smokeview visual output file available in the public GitHub repository "Mallee/Mulga FDI 100 BAL29"

which is mentioned in the Supplementary Materials [59]. The majority of firebrands tend to land along the vegetation's edge, while only a limited number ascend with the convective column, with a maximum release height of 3 m. The wind speed at a height of 3 m is consistently lower across all Fire Danger Index (FDI) scenarios, as shown in Figure 3. This is in contrast with wind speeds observed at higher elevations, such as those encountered in forest fires, where the average height is approximately 40 m, as discussed in our prior study [24]. Consequently, in Mallee/Mulga fires, where the vegetation height is lower, the wind lacks significant momentum to transport a substantial number of firebrands compared to forest fires.

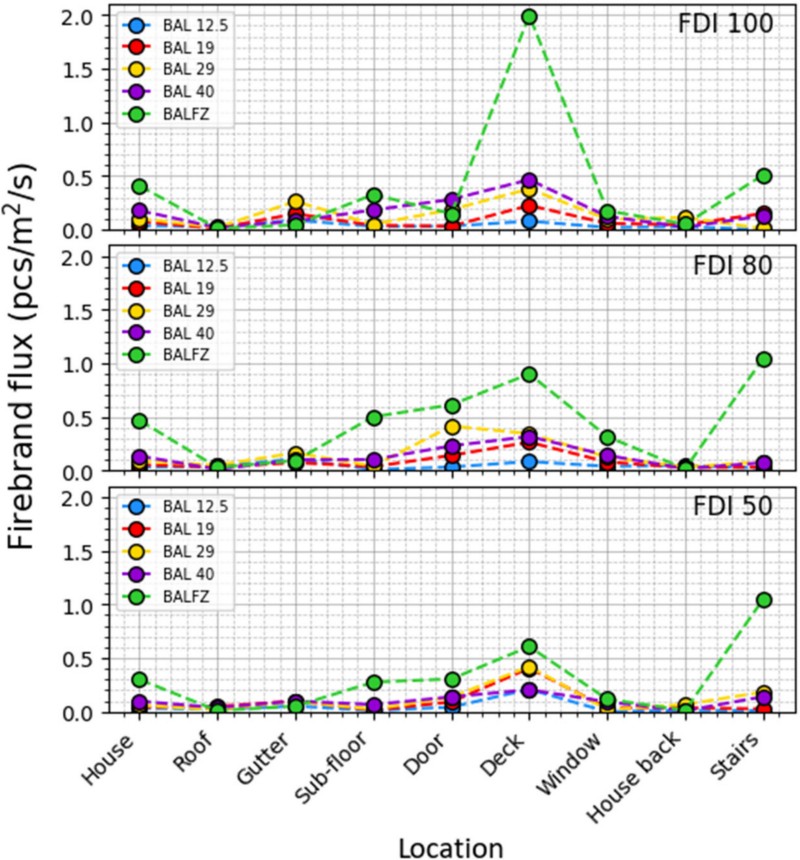

**Figure 7.** Firebrand flux on the strategic locations of the house for FDI 100, 80, and 50 for different BALs. The first tick of the *X*-axis exhibits the total firebrand flux on the house.

In the context of FDI 80, the lower three BALs (12.5, 19, and 29) demonstrate the highest firebrand flux on doors, decks, and windows. However, BAL 40 and BAL FZ deviate from this pattern; for instance, firebrand fluxes at the door and deck for BAL 40 are less than those for BAL 29. Furthermore, the uppermost strategic locations on the house, such as the roof and gutter, exhibit relatively lower firebrand flux compared to the deck and door, particularly beyond BAL 12.5. This suggests that reducing the gap between the house and vegetation increases the vulnerability of front-facing locations compared to upper locations. The flat trajectories of firebrands contribute to the higher firebrand flux at the deck, and at the door area, similar to the observations in FDI 100. However, the FDI 80 fire event shows an even lower overall firebrand flux impact on the house compared to FDI 100.

The lower fire intensity of FDI 50 corresponds to a reduced number of firebrand-generation rates compared to FDIs 80 and 100. Consequently, the total firebrand flux on the house continues to reduce further in comparison to FDI 100 and 80. Remarkably, the deck consistently experiences the highest firebrand flux, whereas the lowest levels are recorded

at the back of the house. In contrast to other fire events (FDI 100 and 80), firebrand fluxes exhibit minimal variation across multiple locations for different BALs under FDI 50.

### 3.3. Radiative Heat Flux on the House

In this study, we examined both the maximum and average radiative heat flux reaching houses. According to the standards defined in AS1530.4-2005 [60], the time it takes for housing materials, such as timber, to catch fire, as well as the impact on materials within a house, can vary depending on the intensity of radiative heat and the duration of exposure. For example, timber can ignite in as little as 10 s when exposed to a radiative heat flux of 55 kW/m$^2$, while it takes 20 s at 45 kW/m$^2$ [21,60]. Similarly, exposure to radiative heat flux levels of 4 kW/m$^2$ and 10 kW/m$^2$ can pose risks/pains to residents in a house, with ignition times ranging from 3 to 10 to 20 s, respectively. By analysing the time-series data of radiative heat flux, we observed fluctuations in these heat flux levels, with the highest radiative heat flux lasting for a shorter duration compared to the average radiative heat flux. To account for higher-risk scenarios, we considered the maximum radiative heat flux, and the average radiative heat flux is used to generalize the risk to houses.

The gutter area experiences the highest radiative heat flux, reaching approximately 120 kW/m$^2$ at BAL FZ. This phenomenon is depicted in Figure 8a (top), where a clear inverse relationship between the maximum radiative heat flux on the house and the distance from the vegetation edge is evident. The most significant heat fluxes are directed towards the gutter, deck, and front wall—the regions directly exposed to the flames. The variation between the maximum and average radiative heat flux for both the gutter and deck is relatively smaller compared to other locations. The elevated radiative heat flux at the gutter is attributed to its closer proximity to the tilting flame, unlike the deck area. This phenomenon is explained using the inverse square law (the intensity decreases proportionally to the square of the distance from the source) of radiative heat flux. Conversely, the back wall consistently receives the lowest radiation across all BALs. This trend extends to the average radiative heat fluxes, as displayed in Figure 8b (top), with substantially lower values. The observed highest average radiative heat flux remains under 30 kW/m$^2$.

Figure 8a (middle) illustrates the maximum radiative heat fluxes on the house during a fire at FDI 80. Similar to the FDI 100 scenario, an upward trend in maximum radiative heat flux with increasing BAL is noticeable at every location. Remarkably, the gutter and deck encounter the most intense radiative heat fluxes, distinct from less-flame-exposed areas such as the back wall. Windows and the front wall, situated at comparable distances from the fire front, exhibit similar radiative heat fluxes from BAL 12.5 to 40. Enhanced flame angles on the house's front side are responsible for the heightened heat fluxes detected at these strategic positions. These trends are illustrated in Figure 8b (middle), wherein the highest average radiative heat flux is approximately 68% lower than the corresponding maximum radiative heat flux (Figure 8a (middle)) for BAL 12.5. This discrepancy reaches about 73% for BAL FZ. Nevertheless, the variation in average radiative heat flux across strategic locations closely resembles that of the maximum radiative heat flux.

Figure 8a,b (bottom) unveils the maximum and average radiative heat fluxes for FDI 50, with the gutter experiencing the most elevated values. Contrary to FDI 100 and 80 scenarios, where the deck area receives the highest radiative heat flux, FDI 50 showcases negligible differences between the radiative heat fluxes at the gutter and the deck. Following the trend of FDI 100 and 80, the back wall consistently endures the least radiation. Correspondingly, the average radiative heat flux under FDI 50 remains lower than that of FDI 100 and 80. The deck area shows the highest average radiative heat flux, followed by the gutter, windows, and front wall. However, all these values remain under 20 kW/m$^2$.

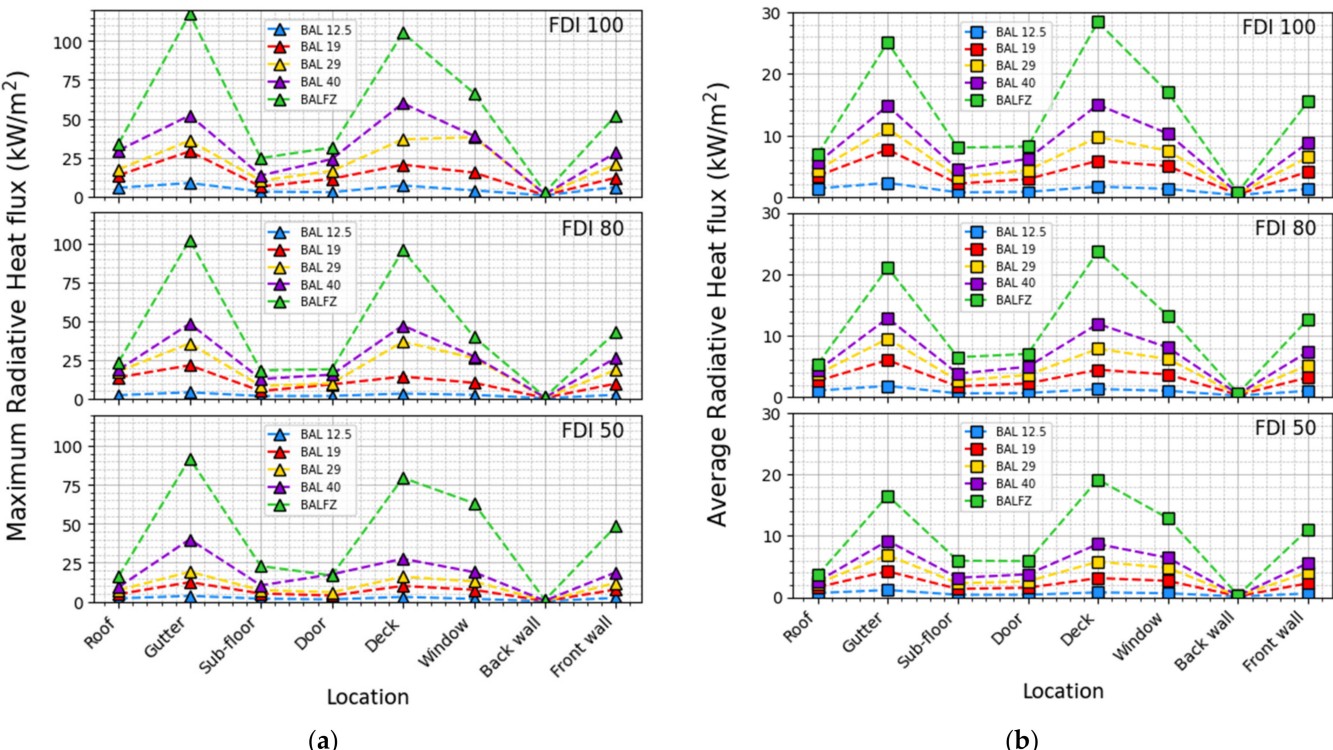

**Figure 8.** (**a**) Maximum and (**b**) average radiative heat flux received by the house at FDI 100, 80, and 50 in BAL 12.5 to FZ.

### 3.4. Convective Heat Flux on the House

Figure 9 illustrates the highest and average convective heat fluxes experienced by the houses at FDI levels of 100, 80, and 50. Notably, the observed convective heat fluxes across distinct strategic positions follow a similar pattern as the radiative heat fluxes for each BAL and the values are lower in magnitude. For instance, in the case of FDI 100, the maximum radiative heat flux is approximately 3 to 19 times greater compared to the maximum convective heat flux from BAL 12.5 to FZ.

The difference between maximum convective heat flux in FDI 80 and 100 is only 3.09 kW/m$^2$ at the window. This value is 2.78 kW/m$^2$ at the gutter, 1.36 kW/m$^2$ at the sub-floor, and less than 1 kW/m$^2$ for all other locations. Investigating the ratio between maximum convective and radiative heat fluxes at strategic locations (specifically, FDI 80) across different BALs (BAL 12.5 to BAL FZ) reveals varying ratios of 73%, 34%, 26%, 22%, and 20%, implying the difference between maximum convective and radiative heat fluxes is reduced with increasing the BAL. It underscores the need to account for convective heat flux, especially at higher BALs, when calculating total heat flux on the houses in WUI. The average convective heat flux pattern in strategic locations closely mirrors the pattern of maximum convective heat flux observed at FDI 80. The proportion of average convective heat flux to maximum convective heat flux at the gutter (highest heat flux receiving location) varies only from 21% to 28% for BAL 12.5 to BAL 40.

Differences in maximum convective heat flux between FDI 50 and FDI 80 show a range of 1.18 to 2.92 kW/m$^2$ at the gutter, the most vulnerable location to heat flux. Similar to the FDI 80 scenario, the difference between the convective heat flux and the radiative heat flux decreases with the BALs. For instance, at the gutter area, the ratio of maximum convective heat flux (as shown in Figure 9a) to maximum radiative heat flux (as illustrated in Figure 8a) varies from 57% to 10% across BALs of 12.5 to BAL FZ. This behaviour is similar to the average convective and average radiative heat fluxes too. Importantly, the average convective heat fluxes remain consistently below 2 kW/m$^2$ across all simulated scenarios as exhibited in Figure 9b.

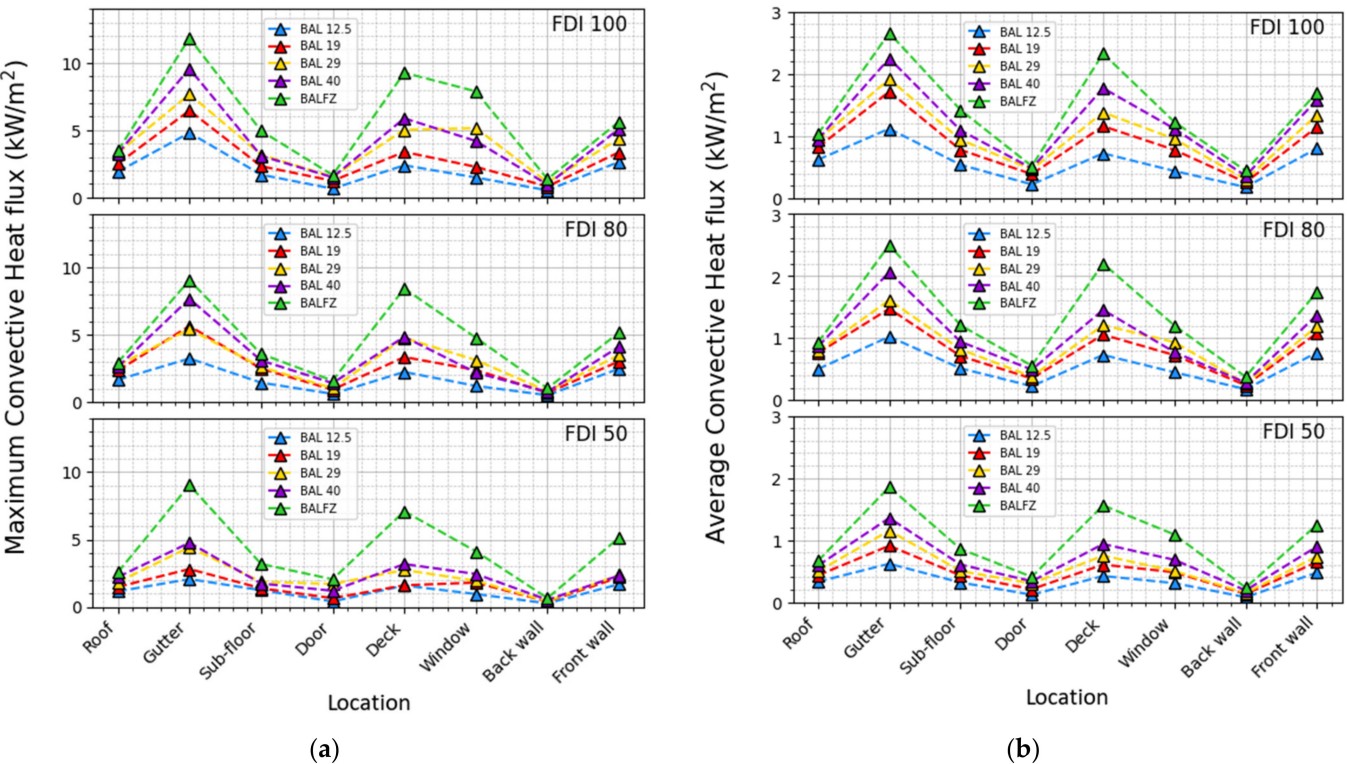

**Figure 9.** (**a**) Maximum and (**b**) average convective heat flux received by the house at FDI 100, 80, and 50 in BAL 12.5 to FZ.

### 3.5. Summary of Firebrand Flux and Radiative Heat Flux According to FDIs and BALs

Figure 10a illustrates the total firebrand flux on the house in relation to the FDI. The total firebrand flux increases as both FDI and the BAL rise, except for BAL FZ. The firebrand flux is slightly higher in BAL FZ for FDI 80 than 100, possibly due to the complex and instantaneous nature of the firebrand landing in the flame zone. Averaging the total firebrand fluxes yields values of 0.161, 0.158, and 0.107 pcs/m$^2$/s for FDI 100, 80, and 50, respectively. The difference in average firebrand flux between FDI 80 and 100 is 2.3%, while the difference between FDI 50 and 80 is 32.2%. When excluding BAL FZ, the average firebrand fluxes are approximately 0.099, 0.079, and 0.058 pcs/m$^2$/s for FDI 100, 80, and 50, respectively. The percentage difference between FDI 80 and 100 is 19.8%, and between FDI 50 and 80, it is 26.8%, limited to BALs from 12.5 to 40.

In Figure 10b, variations in maximum and average radiative heat flux, maximum convective heat flux, and average convective heat flux are summarized for FDI 100, 80, and 50 with respect to different BALs. All examined radiative and convective heat fluxes exhibit an upward trend with increasing FDI and BAL, with exceptions observed in the maximum convective heat flux of BAL 19 and 29 for FDI 80. Overall, the average convective heat flux shows an increasing trend with BAL. The percentage differences were computed for the average of each parameter between FDI 80 to 100 and 50 to 80 for each case. For maximum radiative heat flux, these values are 15.5% (FDI 80–100) to 21.7% (FDI 50–80). This becomes 16.6% (FDI 80–100) and 24.5% (FDI 50–80) for average radiative heat flux. The percentage differences are 22.9% (FDI 80–100) and 25.6% (FDI 50–80) for maximum convective heat flux and 10.2% (FDI 80–100) and 32.1% (FDI 50–80) for the average convective heat flux. Notably, it displays that the percentage differences of all the heat fluxes become higher from FDI 50 to 80 than from FDI 80 to 100.

Based on the firebrand flux results, it is advisable to prioritize the implementation of construction requirements for the foremost areas of houses, such as decks and windows, compared to the upper locations such as gutters or roofs. The back of a house is comparatively at a lower risk of firebrand attacks. However, concerning radiative and

convective heat flux, both gutters and decks require more attention to prevent ignition. By examining the magnitudes of heat fluxes at these locations under various fire conditions, fire experts may gain insights into selecting suitable construction materials for these components of houses.

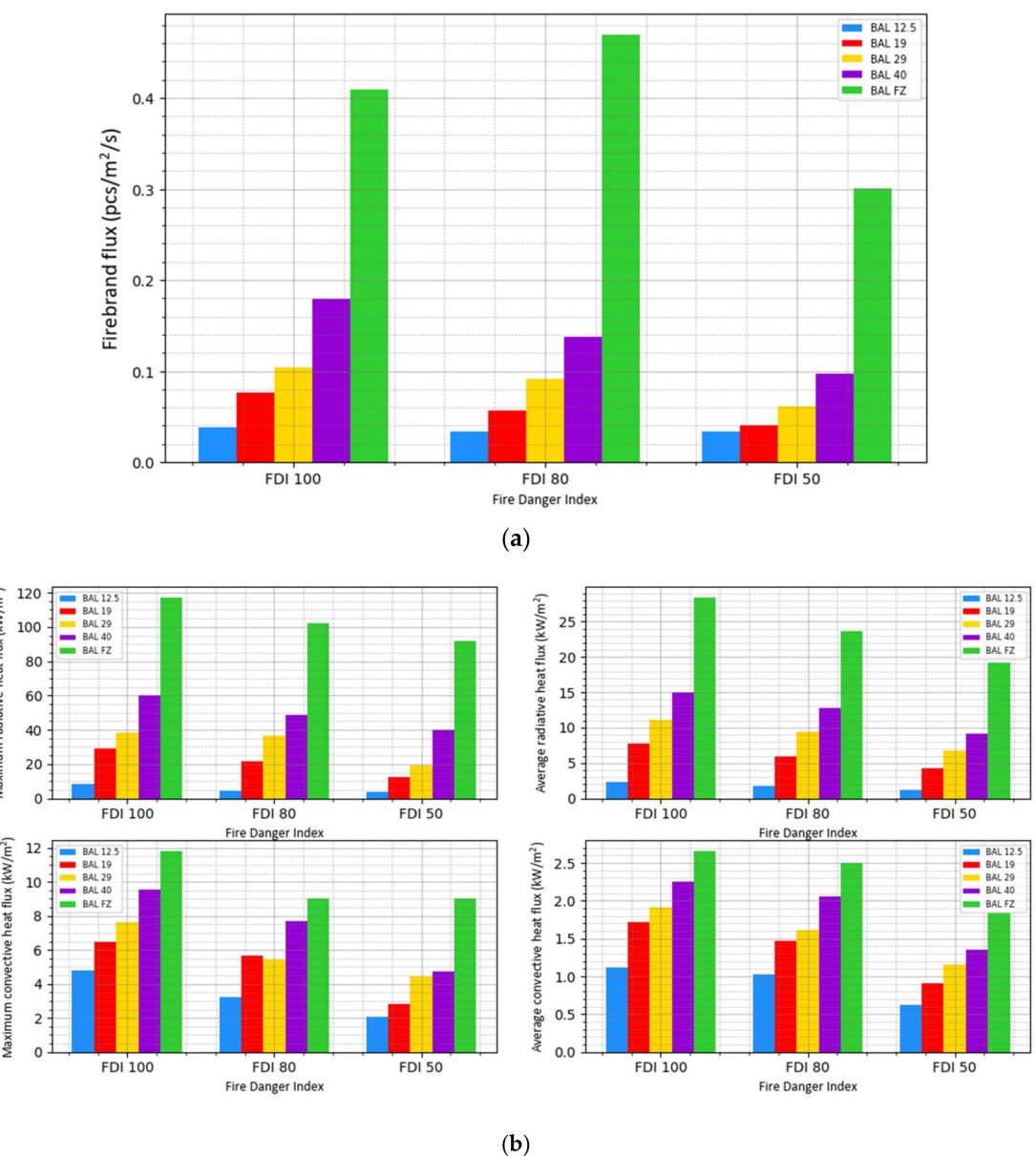

**Figure 10.** The (**a**) total firebrand flux on the house as per FDIs and BAL12.5–BAL FZ, and (**b**) comparison of maximum radiative heat flux, average radiative heat flux, maximum convective heat flux, and average convective heat flux for FDIs and each BAL.

### 3.6. Developing a Correlation between Radiative Heat Flux and the Firebrand Flux

Aiming to establish quantifiable measures, we correlate firebrand flux to the radiative heat flux, as presented in Figure 11. As there is no substantial distinction emerging between firebrand fluxes and the radiative heat fluxes across FDIs and relative BALs, we combined results from FDI 100, 80, and 50 to construct a single correlation between firebrand flux and radiative heat flux. Additionally, as shown in Table 1, the BAL boundaries for FDI 100, 80, and 50 align with those provided in AS3959, making it sensible to consider the combined results. Our analysis encompasses both maximum and average radiative heat fluxes, revealing an exponential relationship with the firebrand flux experienced by the

structures. In Figure 11a,c, the firebrand flux is plotted against the maximum and average radiative heat flux for BAL 12.5 to FZ. Although the BAL FZ seems an outlier, the results agree with the exponential relationship with an $R^2$ of 0.96.

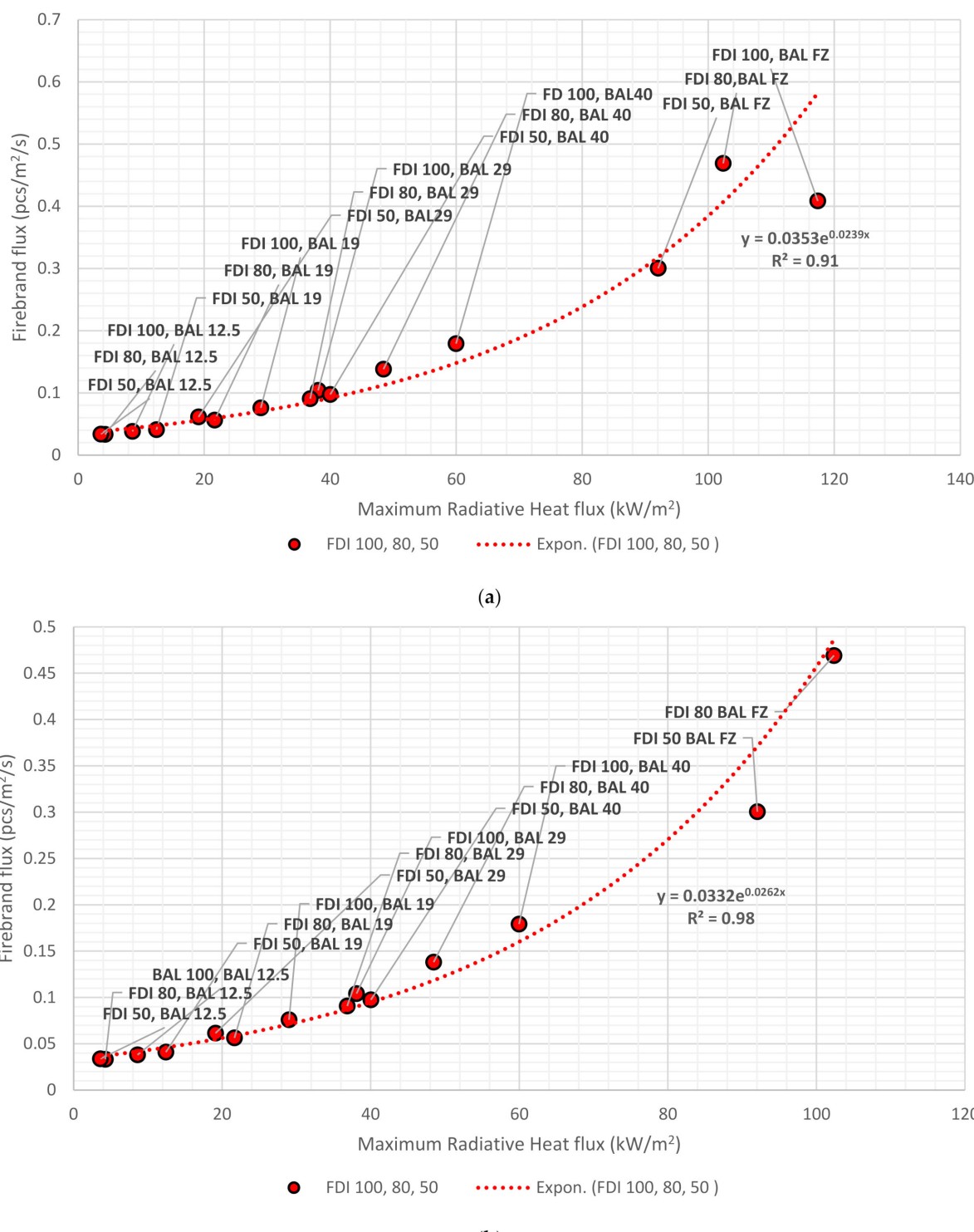

**Figure 11.** *Cont.*

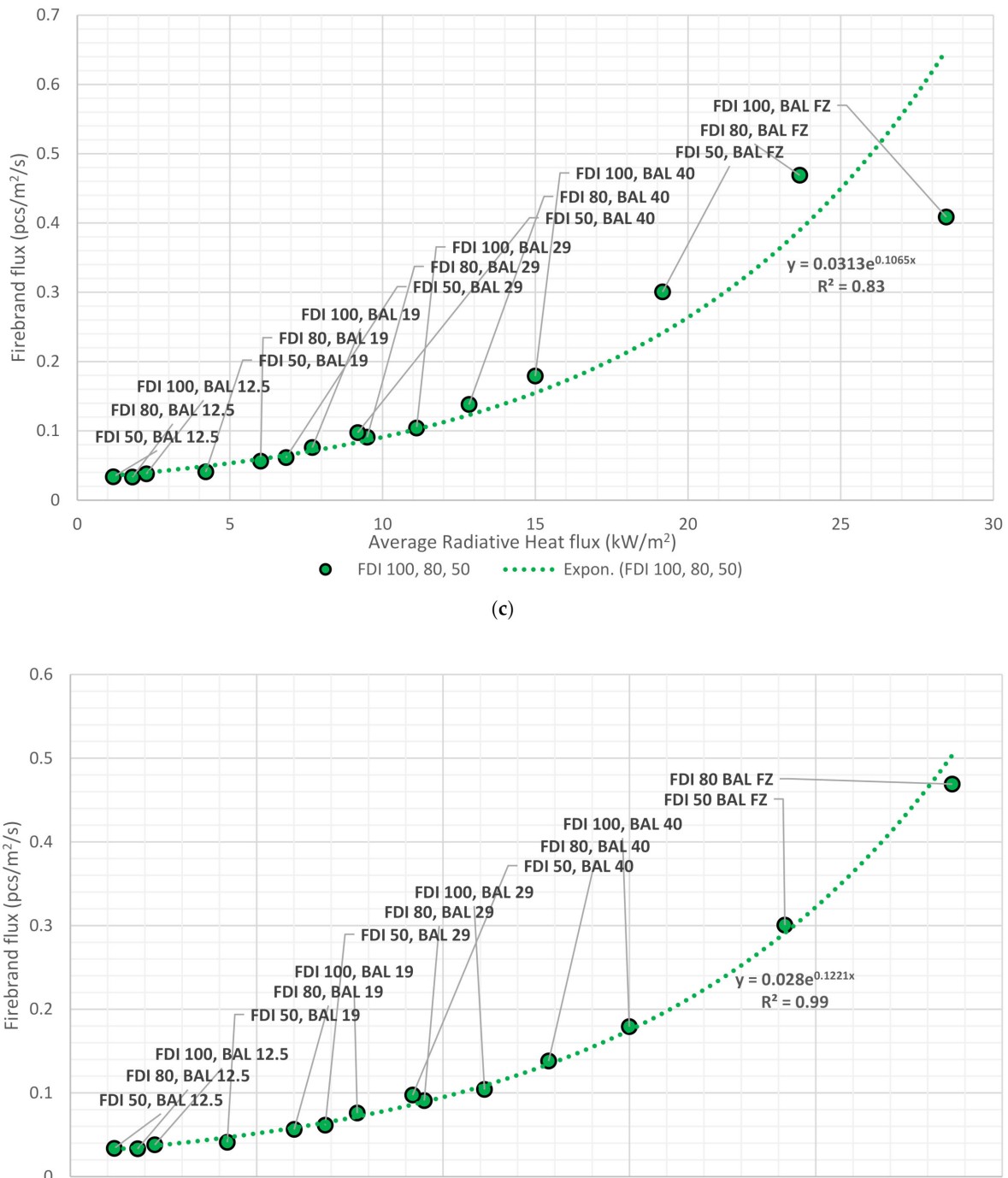

**Figure 11.** The firebrand flux against (**a**) maximum radiative heat flux including FDI 100 BAL FZ, (**b**) maximum radiative heat flux excluding FDI 100 BAL FZ, (**c**) average radiative heat flux including FDI 100 BAL FZ, and (**d**) average radiative heat flux excluding FDI 100 BAL FZ.

Similarly, Figure 11b,d from BAL 12.5 to BAL 40 (excluding BAL FZ) validate the exponential relationship with an $R^2$ over 0.98. In FDI 100 BAL FZ, the maximum and average radiative heat flux are significantly higher and detached from the trend of radiative heat flux of other events. This could be due to the gap between the vegetation and the

house which is minimal and creates a complex environment compared to the other fire events. Therefore, we excluded FDI 100 BAL FZ in Figure 11b,d to investigate the accuracy of using exponential relationships to quantify wildfire risk. Grouping data points by FDIs and BALs sometimes exhibits that the radiative heat flux at a given BAL is lower than that of the preceding BAL in FDI 80 and 100. For example, BAL 29 of FDI 50 is lower than BAL 19 of FDI 100 and 80. This difference narrows at lower BALs and expands with higher BALs, indicating that firebrand attacks on structures exponentially increase when the distance between the vegetation and structure becomes shorter. Depending on the requirements, the developed correlations of firebrand flux with maximum and average radiative heat fluxes serve as potent tools for estimating firebrand risk in WUI. For example, using the correlation presented in Figure 11a, when the maximum radiative heat flux is 25 kW/m$^2$, the firebrand flux on the house will be 0.063 pcs/m$^2$/s. Meanwhile, an average radiative heat flux of 25 kW/m$^2$ (in Figure 11c) yields a value of 0.449 pcs/m$^2$/s. As expected, a significant difference is observed in firebrand flux landing depending on whether maximum or average radiative heat flux is employed as the basis for assessment.

### 3.7. Limitations and Uncertainties

There are several limitations and uncertainties involved in the calibration of firebrand generation to select similar vegetation species, wind, and fuel moisture content measurements from the experiments and use them to develop correlations. The assumption of a similar firebrand-generation rate from similar species brings uncertainty. Among similar species, there may be several differences in the structure of the plant, the nature of the component (barks, twigs, leaves, cones), their behaviours in different climate and weather conditions, plant integrity, etc. However, accounting for these factors in firebrand generation is yet to be fully explored and quantified. The morphologies of the firebrands could vary from one species to another due to a number of factors such as the thickness and stiffness of barks, the flexural rigidity of stems, structural integrity of twigs on exposure to heat, weight, and the size of tree leaves in the array of the canopy. Accounting for all these influential factors increases the number of variables and the complexity of simulations. Simplifying these effects and assuming the same for similar species adds an unquantified uncertainty in this work.

The number of trees in the same species burnt varying FMC [53] and wind speed [54] is only three for each, and these data were taken to develop correlations to find the effect of FMC and wind. Due to the smaller number of data points, there is an uncertainty to these developed correlations which affects the calculated firebrand-generation number in the FDIs. Generating many data points by conducting an experiment burning a large number of trees while varying FMC and wind speed in each series can reduce the uncertainties of developing the correlations. Our assumption of 100% fuel consumption (based on AS3959 [21]) adds uncertainty to the firebrand and radiative heat flux which could be significant upon the actual fuel consumption of Mallee/Mulga vegetation fires.

## 4. Conclusions

In this study, we simulated the firebrand flux on modelled structure in Mallee/Mulga (*Acacia* dominant) WUI fires. The vegetation was modelled as outlined in AS3959 accounting for properties such as height, fuel load, and thermo-physical properties. This structure was positioned based on the respective BALs. The firelines were prescribed with the intensities that were computed according to the severity of the fires given by the FDIs. The firefront was set up to match the radiative heat flux on the structure following the empirical model in AS3959. A comprehensive analysis was carried out focusing on crucial areas where firebrands frequently landed on the housing components for each Fire Danger Index (FDI). By strategically assessing both maximum and average radiative and convective heat fluxes, the structural components' vulnerabilities were assessed. The quantification encompassed computing the average firebrand flux and heat flux (both maximum and average) for each FDI, along with evaluating the percentage variations between FDI 100

and 80 and FDI 80 and 50. The findings revealed the differences in firebrand flux and heat fluxes between FDI 80 and 50 consistently exceed those between FDI 100 and 80. Therefore, we consider all FDIs together to develop one correlation to quantify the firebrand flux against the radiative heat flux. It was found that the radiative heat flux and the firebrand flux of Mallee/Mulga vegetation follow an exponential relationship. The result emphasizes the possibility of quantification of firebrand flux in relation to the radiative heat flux and provides a recommendation to consider firebrand quantification for building standards such as AS3959. The methodology adopted in this study can be utilized for other forest types with different house designs under different atmospheric conditions across different jurisdictions.

**Supplementary Materials:** The following smokeview visuals are available online at https://github.com/Amilwickz/PhD-Thesis_Mallee_Mulga-FDI-100-BAL-29-Smokeview.

**Author Contributions:** Conceptualization, K.M. and A.W.; methodology, K.M. and A.W.; software, A.W.; formal analysis, A.W., K.M., N.K. and A.F; investigation, A.W., K.M., N.K. and A.F.; resources, K.M.; data curation, A.W.; writing—original draft preparation, A.W.; writing—review and editing, N.K., A.F. and K.M.; visualization, A.W.; supervision, K.M., N.K. and A.F.; project administration, K.M.; funding acquisition, K.M. All authors have read and agreed to the published version of the manuscript.

**Funding:** Alexander Filkov was funded by the ARC DP210102540 'Understanding the Origin and Development of Extreme and Mega Bushfires: Merging Fire Fronts.

**Institutional Review Board Statement:** Not applicable.

**Informed Consent Statement:** Not applicable.

**Data Availability Statement:** Data are contained within the article and supplementary materials.

**Acknowledgments:** We would like to acknowledge the Bushfire and Natural Hazard Cooperative Research Centre for the financial support given.

**Conflicts of Interest:** The authors declare no conflict of interest.

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
