# Peer review of "Quantifying Firebrand and Radiative Heat Flux Risk on Structures in Mallee/Mulga-Dominated Wildland–Urban Interface: A Physics-Based Approach"

_fire, doi:10.3390/fire6120466_

Round 1

Reviewer 1 Report

Comments and Suggestions for Authors

Review of “Quantifying Firebrand and Radiative Heat Flux Risk on Structures in Mallee/Mulga-Dominated Wildland Urban Interface: A Physics-Based Approach” by Wickramasinghe et al.

In this paper, the authors have made a quantification of firebrand flux on houses according to the Bushfire Attack Levels in Mallee/Mulga-dominated vegetation at Wildland Urban Interface using physics-based modelling. I found the paper interesting, well-constructed and written, pleasant to read and rich in scientific perspectives, since the methodology adopted in the study can be utilized for other forest types with different house designs and under different atmospheric conditions. I have no specific concerns about the methodology and the analysis of the results. I thus consider that the paper can be accepted with only minor revision.

Minor comments

Lines 12-27. Please, the abstract does not follow the recommendation of the journal of 200 words maximum.

Line 119. The number of the sections is not right. Please, change “3” to “2”. The same for the subsections in lines 131, 155, 196, 218, 242, 333, and 349.

Line 319. Please, confirm the results presented in the column of “total firebrand generation rate”.

Line 365. Please change the number of the section to “3”. The same for the subsections in lines 366, 390, 422, 465, 493, 517, and 555.

Line 580. Please change the number of the section to “4”.

Reviewer 2 Report

Comments and Suggestions for Authors

This study investigates the influence of firebrand attack and radiant heat flux on various components of a house, utilizing FDI and BAL methods. There are several major concerns that I would like the authors to consider/address:

1. In L69-70, authors said “AS3959 considers seven vegetation classes”, while this article only investigated Mallee/Mulga. I suggest authors provide more characteristics of Mallee/Mulga wildfires in the introduction.

2. Wildfires damage housing components may differently depend on the topography. This study

only focused on the BALs for 0-degree slope (L149-150). I suggested to provide more information, why the authors only focused on 0-degree slope. Is it better to include more slopes?

3. The current study lacks adequate discussion of the results. For instance, although the gutters in Figures 7, 8, and 9 differ, the author focuses more on describing the outcomes without delving into the underlying mechanisms. Furthermore, how to comprehensively evaluate the differences between these results to better implement your goals (L115-118).

Comments on the Quality of English Language

The quality of English is fine.

Reviewer 3 Report

Comments and Suggestions for Authors

In this paper,he findings reveal an increasing firebrand flux with higher BAL values across all FDIs, with a greater percentage difference observed between FDIs 50-80 compared to FDIs 80-100. Furthermore, an exponential relationship is observed between radiative heat flux and firebrand flux. This research contributes to a better understanding of firebrand dynamics and informs the development of effective strategies to enhance the resilience of structures to improve AS3959.

1. The references in this article need to be updated appropriately.

Comments on the Quality of English Language

NO
